# Dynamics of Large-Scale Solar-Wind Streams Obtained by the Double Superposed Epoch Analysis: 5. Influence of the Solar Activity Decrease

Yuri I. Yermolaev *, Irina G. Lodkina, Alexander A. Khokhlachev, Michael Yu. Yermolaev, Maria O. Riazantseva, Liudmila S. Rakhmanova, Natalia L. Borodkova, Olga V. Sapunova and Anastasiia V. Moskaleva

Space Research Institute, Russian Academy of Sciences, 117997 Moscow, Russia
* Correspondence: yermol@iki.rssi.ru

**Abstract:** In solar cycles 23–24, solar activity noticeably decreased and, as a result, solar wind parameters decreased. Based on the measurements of the OMNI base for the period 1976–2019, the time profiles of the main solar wind parameters and magnetospheric indices for the main interplanetary drivers of magnetospheric disturbances (solar wind types CIR. Sheath, ejecta and MC) are studied using the double superposed epoch method. The main task of the research is to compare time profiles for the epoch of high solar activity at 21–22 SC and the epoch of low activity at 23–24 SC. The following results were obtained. (1) The analysis did not show a statistically significant change in driver durations during the epoch of minimum. (2) The time profiles of all parameters for all types of SW in the epoch of low activity have the same shape as in the epoch of high activity, but locate at lower values of the parameters. (3) In CIR events, the longitude angle of the solar wind flow has a characteristic S shape; but in the epoch of low activity, it varies in a larger range than in the previous epoch.

**Keywords:** solar wind; interplanetary phenomena; interplanetary drivers; solar-terrestrial physics

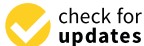



## 1. Introduction

The hot solar corona expanding into interplanetary space forms the solar wind (SW). The inhomogeneity and nonstationarity of the solar corona led to the formation of large-scale phenomena (structures) in the solar wind (see, for example, the reviews [1–6] and references therein). Usually, quasi-stationary and perturbed types of phenomena are distinguished. The former include phenomena that are associated with long-lived solar structures: slow streams from the region of coronal streamers and fast streams from coronal holes. Since the boundary of the reorientation of the coronal magnetic field (neutral line) is located in the region of the streamer belt, this boundary is projected into the slow SW in the form of the so-called Heliospheric Current Sheet (HCS). Disturbed streams include CME-related phenomena, magnetic clouds and ejecta, the differences between which are higher and more regular interplanetary magnetic field (IMF) in clouds compared to ejecta. When fast flows from coronal holes and fast MC/ejecta interact with slower preceding SW flows, compression regions, corotating interaction region (CIR) and sheath, respectively, are formed in the interplanetary medium. When the speed of fast outflows from coronal holes and fast MC/ejecta exceeds the speed of previous SW outflows by the magnitude of the magnetosonic speed, a shock is formed at the leading edge of CIR and sheath. If the speed of trailing edges of MC/ejecta or fast flows from coronal holes is less than the speed of preceding SW flows, a sparse region is formed, the so-called Rarefied stream. There are a number of catalogs (or lists) that include both distinct types of SW (see, for example, CIR [7], sheath + MC/ejecta [8,9], the Rarefied region [10], as well as the complete set of SW types by [11]. Since the disturbed SW types play the main role in the transfer of disturbances from the Sun to the Earth, then, as a rule, all such catalogs/lists of phenomena

are created and developed for the problems of solar-terrestrial physics and space weather. Nevertheless, they provide a sufficient opportunity for studying the physics of various phenomena in the SW.

SW research began shortly after the beginning of the space age [12,13], and systematic SW measurements cover 20–25 solar cycles (SCs) (see, for example, interplanetary space measurements of OMNI data [14]). During this time, the Sun passed the epoch of maximum and entered the epoch of minimum at the beginning of SC 23 [15–17]. At this phase, in 23–24 SCs, the fraction of CIR in SW slightly changed, while the fractions of MC/ejecta and sheath significantly fell compared to 21–22 SCs [18,19]. In addition, in SC 23–24, all the main plasma parameters and IMF dropped by 20–40% in all types of SW and in all phases of solar cycles [18–20].

In our paper [21], we started a series of works on the statistical study of interplanetary drivers [22–24]. In the first article, we studied the time profiles of the main parameters of SW and IMF for the eight usual sequences of SW phenomena: (1) SW/CIR/SW, (2) SW/IS/CIR/SW, (3) SW/ejecta/SW, (4) SW/sheath/ejecta/SW, (5) SW/IS/sheath/ejecta/SW, (6) SW/MC/SW, (7) SW/sheath/MC/SW, and (8) SW/IS/sheath/MC/SW (where SW means undisturbed SW and IS means interplanetary shock) for 1976–2000. These data mainly include the epoch of maximum: the full 21–22 SC (1976–1996) and the beginning of 23 SC (1997–2000). In this paper, we similarly recalculate time profiles at SC 21–22, examine the time profiles of the same parameters at the epoch of minimum at SC 23–24 (1997–2019), and look for possible differences in time profiles at the epoch of maximum and epoch of minimum.

This paper has the following structure. Section 2 describes the data used and the methodology for their analysis. The results obtained are presented in Section 3. Section 4 is devoted to a discussion of the results obtained and the conclusions of this work.

## 2. Data and Methods

In this work, we use the same sources of information as in the previous paper [21]: (1) hourly data of the OMNI database parameters for 1976–2019 (https://spdf.gsfc.nasa.gov/pub/data/omni/low_res_omni [14], accessed on 1 November 2021), and (2) intervals of different SW types in the catalog of large-scale phenomena (http://www.iki.rssi.ru/pub/omni [11], accessed on 1 January 2022), created on the basis of the OMNI database.

In this paper, we present the results in two forms: figures and tables. The figures show the average time profiles of 21 solar wind parameters and magnetospheric indices calculated separately for the phases of high (1976–1996) and low (1997–2019) solar activity and separately for 8 characteristic sequences of disturbed SW types: (1) SW/CIR/SW, (2) SW/IS/CIR/SW, (3) SW/ejecta/SW, (4) SW/sheath/ejecta/SW, (5) SW/IS/sheath/ejecta/SW, (6) SW/MC/SW, (7) SW/sheath/MC/SW, and (8) SW/IS/sheath/MC/SW. The tables present the averaged values of the same parameters for the indicated SW types, as well as the average values at the last undisturbed SW point (SW [a]) and the first undisturbed point (SW [b]) adjoining the corresponding sequences of disturbed SW types.

To calculate the average time profile, this paper uses the method of double superposed epoch analysis (DSEA) [25] similarly to how it was done in the previous article [21]. The method involves rescaling (proportional increasing/decreasing time between points) the duration of the interval for all SW types in such a manner that, respectively, the beginnings and ends of all intervals of a selected type coincide. Similar methods of profile analysis were used in the papers by [26–28]. Since one of the key parameters of the DSEA method is the determination and use of average interval durations in data processing, we analyzed the average durations of various disturbed SW types (CIR, sheath, ejecta and MC) separately for 2 time intervals 21–22 and 23–24 SCs. With the existing spread of durations, we were unable to find statistically significant differences in durations for the two intervals, and in further analysis we used the same durations for both intervals that were used in previous paper [21]: 20 h for CIR, 25 h for Ejecta and MC, 14 h for Sheath before Ejecta, and 10 h for Sheath before MC.

Variations in the parameters of the solar wind (even in its separate type) are quite large and led to large standard deviations, σ. To compare the parameters averaged over the intervals 21–22 and 23–24 SC, it is necessary to evaluate the reliability of the obtained average values. For this purpose, a statistical error was calculated, equal to the standard deviation divided by the square root of the number of points indicated in the corresponding columns of the tables. Since the number of events indicated in the figure captions ranges from a few to several hundred, and the number of points is up to several thousand, the statistical error (except for some sequences associated with MCs) turns out to be 1.5–2 orders of magnitude smaller than the standard deviation, and the differences between parameters averaged over the intervals of high and low solar activity have a high statistical significance.

It should be noted that the indicated number of different SW events in different epochs of solar activity cannot be directly interpreted as the prevalence of these events due to the lack of measurements over a large number of time intervals (see, for example, [18]).

## 3. Results

The results of data analysis are presented in the form of eight figures and eight tables. The procedure for obtaining these figures and tables is described in the previous section.

Figures 1–8 present temporal profiles of similar structures for 21–22 (thin lines) and 23–24 (thick lines) SC epochs and show the following parameters:

(a) The solar wind bulk velocity *V* (*black lines*), and the *AE* index (*red lines*);

(b) The ratio of thermal and magnetic pressures *β*, and alpha-particle abundance *Na/Np*;

(c) The ion density *N* and the *Kp* index;

(d) The magnitude of IMF *B* and the dynamic pressure *Pd*;

(e) The solar wind velocity angles: longitude *ϕ* and latitude *θ*;

(f) The proton temperature *Tp* (red), the thermal pressure *Pt (blue)*, and the ratio of measured and expected temperatures T/Texp (black);

(g) The sound and Alfvenic velocities *Vs* and *Va*;

(h) The measured and density-corrected *Dst* and *Dst\** indices.

### 3.1. Variation in CIR Events

Time profiles of 21 SW parameters and magnetospheric indices for CIR events with and without a preceding interplanetary shock wave are presented in Figures 1 and 2, respectively. The average values of these parameters are in Tables 1 and 2.

**Table 1.** Average parameters for sequence **SW/CIR/SW (Figure 1)**.

| Period | 1976–1996 | | | | 1997–2019 | | | |
|---|---|---|---|---|---|---|---|---|
| SW Type | SW [a] | CIR | | SW [b] | SW [a] | CIR | | SW [b] |
| Parameter | < > | < > ± σ | Stat. Err. | < > | < > | < > ± σ | Stat. Err. | < > |
| 1. N, cm$^{-3}$ | 10.9 | 10.9 ± 7.0 | 0.11 | 6.2 | 9.1 | 9.5 ± 6.1 | 0.09 | 5.1 |
| 2. Na/Np (%) | 4.3 | 4.9 ± 2.7 | 0.05 | 5.1 | 2.9 | 3.3 ± 2 | 0.03 | 3.8 |
| 3. V, km/s | 392 | 445 ± 87 | 1.4 | 504 | 382 | 424 ± 86 | 1.2 | 489 |
| 4. Phi, deg | −1.8 | 0.6 ± 3.5 | 0.06 | 2.1 | −1.8 | 0.2 ± 3.5 | 0.05 | 2.1 |
| 5. Theta, deg | 1.4 | 1.1 ± 3.4 | 0.06 | 1.0 | −0.3 | −0.5 ± 2.8 | 0.04 | −0.6 |
| 6. T*10$^{-5}$, K | 0.66 | 1.76 ± 1.23 | 0.020 | 1.76 | 0.6 | 1.36 ± 1.01 | 0.014 | 1.62 |
| 7. T/Texp | 1.3 | 2.23 ± 1.02 | 0.016 | 1.69 | 1.25 | 1.96 ± 0.92 | 0.013 | 1.64 |
| 8. Ey, mV/m | −0.14 | −0.06 ± 1.83 | 0.029 | −0.06 | 0.07 | −0.03 ± 1.57 | 0.022 | −0.03 |
| 9. B, nT | 8.0 | 9.2 ± 3.6 | 0.06 | 7.4 | 6.8 | 7.8 ± 3.1 | 0.04 | 6.6 |
| 10. Bx, nT | −0.4 | −0.4 ± 5.2 | 0.08 | 0.3 | 0.1 | −0.1 ± 4.1 | 0.06 | 0.1 |
| 11. By, nT | 0.6 | 0.4 ± 5.8 | 0.09 | 0.3 | −0.8 | 0 ± 4.8 | 0.07 | −0.1 |
| 12. Bz, nT | 0.3 | 0.1 ± 4.1 | 0.06 | 0.1 | −0.2 | 0.1 ± 3.8 | 0.05 | 0.1 |
| 13. Pt*100, nPa | 0.8 | 2.3 ± 2 | 0.03 | 1.4 | 0.6 | 1.5 ± 1.1 | 0.02 | 1.0 |

**Table 1.** *Cont.*

| Period | 1976–1996 | | | | 1997–2019 | | | |
|---|---|---|---|---|---|---|---|---|
| SW Type | SW [a] | CIR | | SW [b] | SW [a] | CIR | | SW [b] |
| Parameter | < > | < > ± σ | Stat. Err. | < > | < > | < > ± σ | Stat. Err. | < > |
| 14. Pd, nPa | 2.57 | 3.32 ± 1.82 | 0.029 | 2.5 | 2.11 | 2.6 ± 1.31 | 0.018 | 1.88 |
| 15. beta | 0.47 | 0.8 ± 0.8 | 0.013 | 0.77 | 0.48 | 0.71 ± 0.65 | 0.009 | 0.67 |
| 16. DST, nT | −12.7 | −17.4 ± 23.9 | 0.36 | −24.8 | −7.6 | −10 ± 19.2 | 0.27 | −17.0 |
| 17. DST*, nT | −16.2 | −24.7 ± 24.5 | 0.39 | −30 | −9.3 | −14.4 ± 20.1 | 0.28 | −19.0 |
| 18. Kp*10 | 25.5 | 31.2 ± 12.9 | 0.193 | 32.2 | 19.6 | 24.3 ± 12.9 | 0.18 | 24.8 |
| 19. AE | 222 | 282 ± 234 | 3.8 | 289 | 171 | 230 ± 219 | 3.2 | 238 |
| 20. Va, km/s | 53.8 | 62.3 ± 26.7 | 0.43 | 62.8 | 50.3 | 56.4 ± 24.7 | 0.35 | 64.3 |
| 21. Vs, km/s | 52.5 | 65 ± 12.4 | 0.2 | 65.4 | 51.7 | 60.6 ± 11 | 0.15 | 63.9 |

[a] Last SW point before corresponding sequence of disturbed SW types. [b] First SW point after corresponding sequence of disturbed SW types.

**Table 2.** Average parameters for sequence **SW/IS/CIR/SW (Figure 2).**

| Period | 1976–1996 | | | | 1997–2019 | | | |
|---|---|---|---|---|---|---|---|---|
| SW Type | SW [a] | CIR | | SW [b] | SW [a] | CIR | | SW [b] |
| Parameter | < > | < > ± σ | Stat. Err. | < > | < > | < > ± σ | Stat. Err. | < > |
| 1. N, cm$^{-3}$ | 13.0 | 14.0 ± 10.4 | 0.21 | 7.1 | 12.4 | 12.1 ± 8.9 | 0.08 | 5.2 |
| 2. Na/Np (%) | 3.4 | 4.3 ± 2.3 | 0.05 | 4.3 | 2.5 | 4.0 ± 2.4 | 0.03 | 4.2 |
| 3. V, km/s | 376 | 455 ± 97 | 2 | 537 | 361 | 448 ± 94 | 0.9 | 529 |
| 4. Phi, deg | −2.1 | 0.5 ± 4.1 | 0.08 | 2.2 | −2.8 | 0.3 ± 4.3 | 0.04 | 2.4 |
| 5. Theta, deg | 0.9 | 0.3 ± 3.4 | 0.07 | 0.4 | −0.4 | −0.4 ± 3.0 | 0.03 | −0.4 |
| 6. T*10$^{-5}$, K | 0.62 | 1.99 ± 1.64 | 0.033 | 2.37 | 0.54 | 1.75 ± 1.32 | 0.013 | 1.9 |
| 7. T/Texp | 1.36 | 2.28 ± 1.16 | 0.023 | 1.78 | 1.32 | 2.09 ± 1.02 | 0.010 | 1.61 |
| 8. Ey, mV/m | 0.00 | 0.03 ± 2.38 | 0.047 | −0.10 | −0.02 | −0.02 ± 1.9 | 0.018 | −0.02 |
| 9. B, nT | 6.5 | 10.7 ± 4.6 | 0.09 | 8.4 | 6.3 | 9.3 ± 3.4 | 0.03 | 7.0 |
| 10. Bx, nT | −0.4 | 0.1 ± 5.7 | 0.11 | 0.4 | 0.2 | 0.1 ± 4.9 | 0.05 | 0.3 |
| 11. By, nT | 0.0 | −0.1 ± 6.9 | 0.14 | −0.2 | 0.0 | 0.0 ± 5.7 | 0.05 | 0.2 |
| 12. Bz, nT | 0.0 | 0.0 ± 5.2 | 0.10 | 0.4 | 0.1 | 0.1 ± 4.3 | 0.04 | 0.0 |
| 13. Pt*100, nPa | 1.0 | 3.0 ± 2.5 | 0.05 | 1.9 | 0.8 | 2.3 ± 1.8 | 0.02 | 1.3 |
| 14. Pd, nPa | 2.91 | 4.39 ± 2.61 | 0.051 | 3.13 | 2.60 | 3.59 ± 1.98 | 0.019 | 2.30 |
| 15. beta | 0.92 | 0.82 ± 0.86 | 0.018 | 0.84 | 0.65 | 0.74 ± 0.57 | 0.005 | 0.74 |
| 16. DST, nT | −6.8 | −19.0 ± 30.5 | 0.58 | −33.0 | −1.5 | −10.6 ± 23.0 | 0.22 | −23.0 |
| 17.DST*, nT | −10.5 | −30.1 ± 32.2 | 0.63 | −41.3 | −4.0 | −19.0 ± 24.1 | 0.23 | −27.7 |
| 18. Kp*10 | 23.1 | 35.5 ± 14.6 | 0.277 | 36.7 | 18.6 | 29.9 ± 13.1 | 0.125 | 29.8 |
| 19. AE | 193 | 327 ± 265 | 5.4 | 316 | 151 | 280 ± 239 | 2.4 | 302 |
| 20. Va,km/s | 40.7 | 63.7 ± 32.6 | 0.64 | 69.3 | 39.3 | 60.7 ± 25.8 | 0.25 | 64.6 |
| 21. Vs, km/s | 52.0 | 66.9 ± 15.2 | 0.31 | 70.6 | 50.8 | 64.6 ± 13.5 | 0.13 | 66.8 |

[a] Last SW point before corresponding sequence of disturbed SW types. [b] First SW point after corresponding sequence of disturbed SW types.

Figures 1 and 2 and Tables 1 and 2 show that in the epoch of low solar activity (thick lines in the figures), the time profiles in CIR have the same shape, but pass at lower levels than in the epoch of high activity (thin lines). It should be noted that there is a positive latitudinal angle at high activity and a negative angle at low activity.

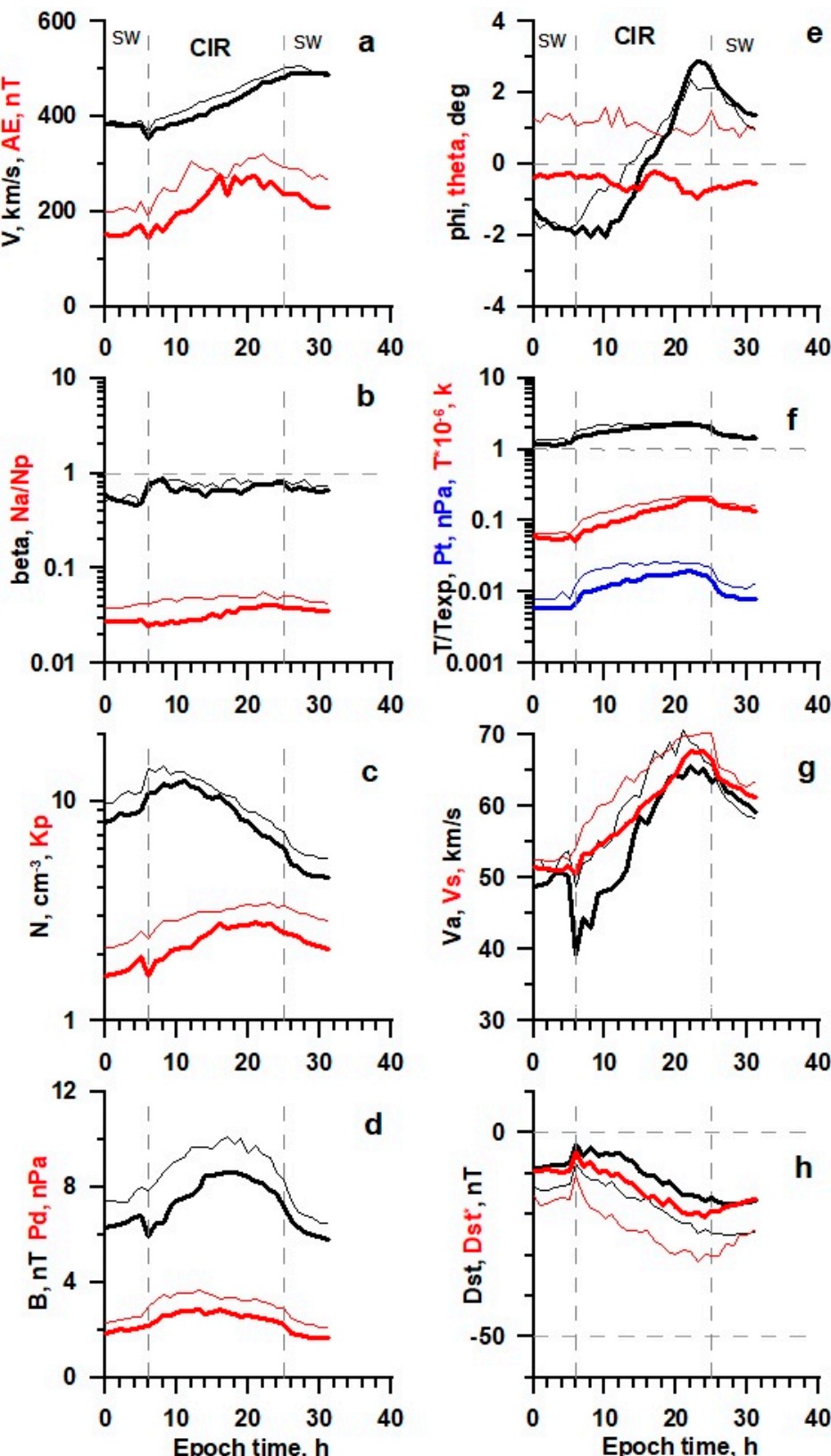

**Figure 1.** The temporal profile of solar wind and IMF parameters (see legend in the text) for CIR obtained by the double superposed epoch analysis for 224 events in 21–22 SCs (thin lines) and 253 events in 23–24 SCs (thick lines).

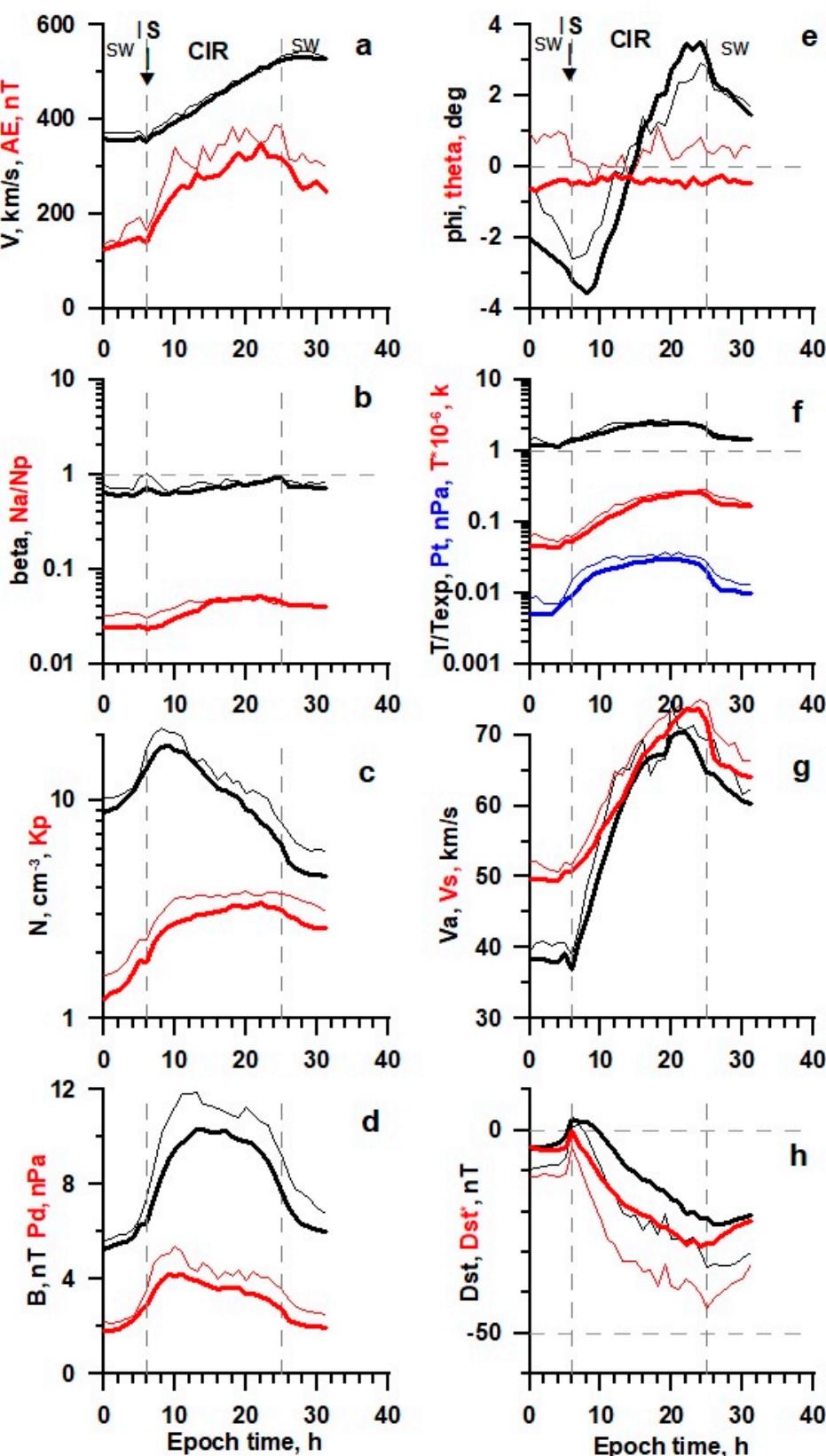

**Figure 2.** The temporal profile of solar wind and IMF parameters (see legend in the text) for IS/CIR obtained by the double superposed epoch analysis for 127 events in 21–22 SCs (thin lines) and 539 events in 23–24 SCs (thick lines).

### 3.2. Variation in Sheath and Ejecta Events

Figures 3–5 show the time profiles for the sequences SW/ejecta/SW, SW/sheath/ejecta/SW, and SW/IS/sheath/ejecta/SW, respectively. The average values of the parameters are presented in Tables 3–5.

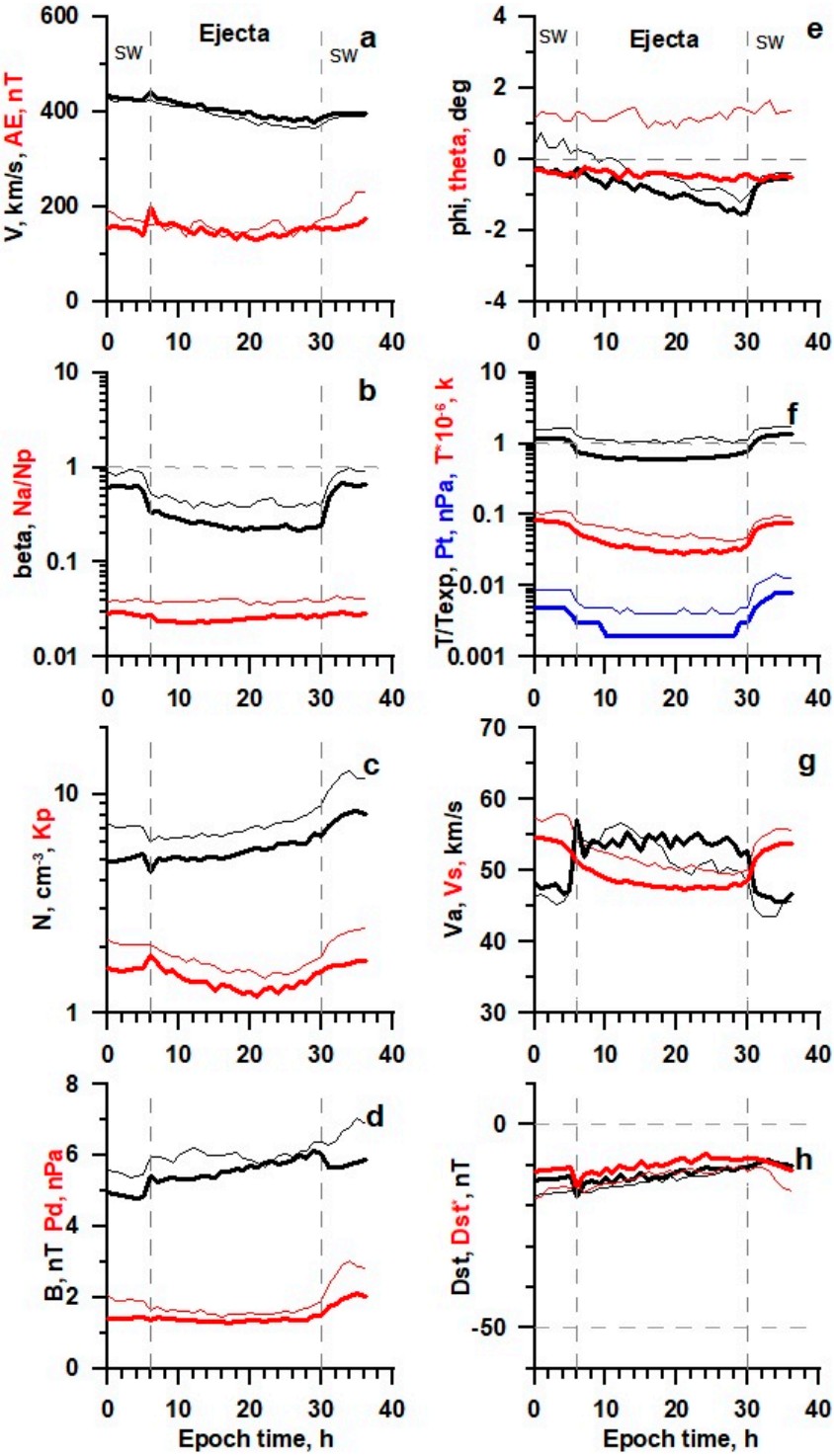

**Figure 3.** The temporal profile of solar wind and IMF parameters (see legend in the text) for Ejecta obtained by the double superposed epoch analysis for 287 events in 21–22 SCs (thin lines) and 491 events in 23–24 SCs (thick lines).

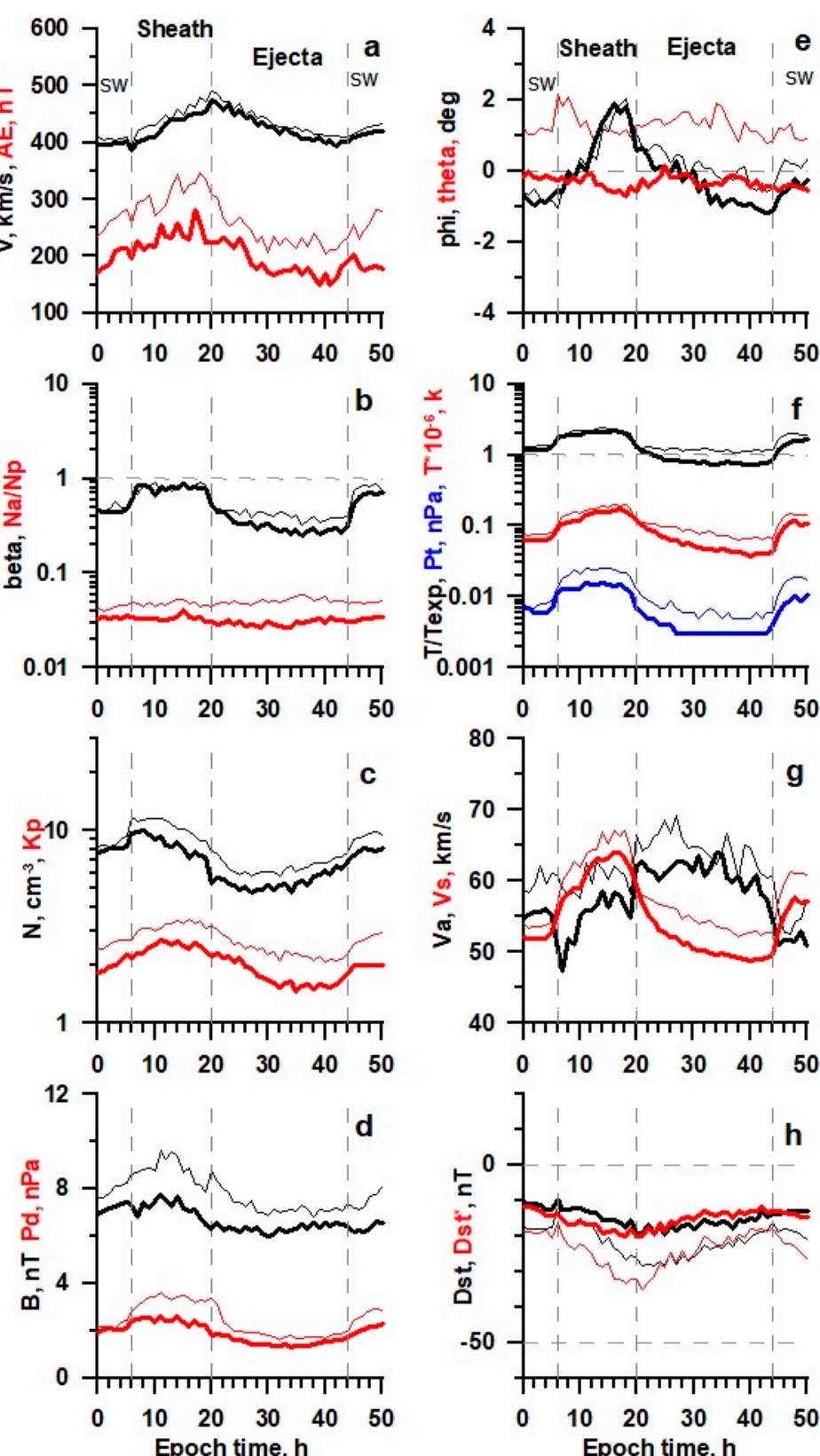

**Figure 4.** The temporal profile of solar wind and IMF parameters (see legend in the text) for Sheath/Ejecta obtained by the double superposed epoch analysis for 188 events in 21–22 SCs (thin lines) and 138 events in 23–24 SCs (thick lines).

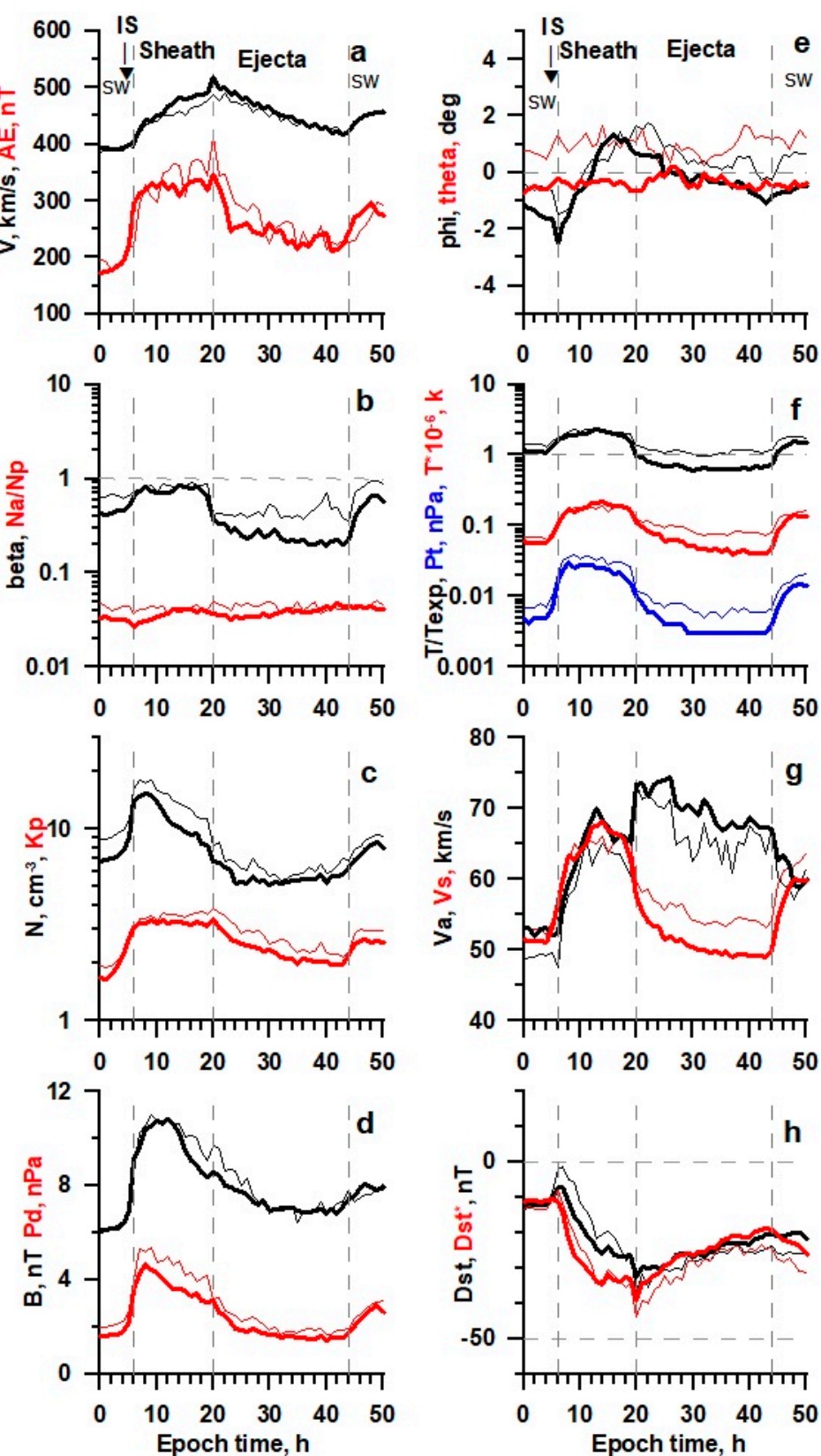

**Figure 5.** The temporal profile of solar wind and IMF parameters (see legend in the text) for IS/Sheath/Ejecta obtained by the double superposed epoch analysis for 110 events in 21–22 SCs (thin lines) and 247 events in 23–24 SCs (thick lines).

**Table 3.** Average parameters for sequence **SW/EJECTA/SW (Figure 3).**

| Period | 1976–1996 | | | | 1997–2019 | | | |
|---|---|---|---|---|---|---|---|---|
| SW Type | SW [a] | EJECTA | | SW [b] | SW [a] | EJECTA | | SW [b] |
| Parameter | < > | < > ± σ | Stat. Err. | < > | < > | < > ± σ | Stat. Err. | < > |
| 1. N, cm$^{-3}$ | 7.1 | 7.0 ± 4.7 | 0.05 | 10.4 | 5.3 | 5.5 ± 3.7 | 0.04 | 7.2 |
| 2. Na/Np (%) | 3.9 | 3.9 ± 2.8 | 0.04 | 4.0 | 2.7 | 2.5 ± 1.9 | 0.02 | 2.9 |
| 3. V, km/s | 422 | 390 ± 74 | 0.9 | 384 | 425 | 401 ± 75 | 0.8 | 394 |
| 4. Phi, deg | 0.2 | −0.5 ± 2.6 | 0.03 | 0.8 | −0.5 | −1.0 ± 2.3 | 0.02 | −0.8 |
| 5. Theta, deg | 1.1 | 1.2 ± 2.8 | 0.03 | 1.3 | −0.4 | −0.4 ± 2.2 | 0.02 | −0.5 |
| 6. T*10$^{-5}$, K | 1.05 | 0.55 ± 0.46 | 0.005 | 0.78 | 0.69 | 0.35 ± 0.31 | 0.003 | 0.60 |
| 7. T/Texp | 1.63 | 1.12 ± 0.76 | 0.009 | 1.60 | 1.05 | 0.65 ± 0.39 | 0.004 | 1.09 |
| 8. Ey, mV/m | −0.15 | −0.04 ± 1.13 | 0.013 | −0.15 | 0.00 | −0.03 ± 1.20 | 0.012 | −0.10 |
| 9. B, nT | 5.5 | 6.0 ± 2.1 | 0.02 | 6.3 | 4.8 | 5.6 ± 2.3 | 0.02 | 5.7 |
| 10. Bx, nT | 0.1 | −0.2 ± 3.6 | 0.04 | 0.1 | 0.2 | 0.1 ± 3.3 | 0.03 | 0.1 |
| 11. By, nT | 0.2 | 0.2 ± 3.9 | 0.04 | 0.0 | −0.3 | −0.1 ± 3.8 | 0.04 | 0.0 |
| 12. Bz, nT | 0.3 | 0.1 ± 2.8 | 0.03 | 0.4 | 0.0 | 0.1 ± 3.0 | 0.03 | 0.2 |
| 13. Pt*100, nPa | 0.9 | 0.5 ± 0.5 | 0.01 | 1.0 | 0.4 | 0.2 ± 0.2 | 0.00 | 0.5 |
| 14. Pd, nPa | 1.90 | 1.63 ± 0.98 | 0.011 | 2.38 | 1.44 | 1.38 ± 0.91 | 0.009 | 1.73 |
| 15. beta | 0.84 | 0.43 ± 0.69 | 0.008 | 0.68 | 0.57 | 0.25 ± 0.36 | 0.004 | 0.48 |
| 16. DST, nT | −16.2 | −13.5 ± 18.6 | 0.2 | −9.4 | −12.6 | −12.2 ± 16.7 | 0.17 | −9.8 |
| 17. DST*, nT | −15.6 | −12.5 ± 18.7 | 0.21 | −10.6 | −10.5 | −9.7 ± 17.5 | 0.18 | −8.2 |
| 18. Kp*10 | 20.5 | 16.6 ± 11.0 | 0.119 | 20.8 | 16.08 | 13.9 ± 11.8 | 0.118 | 16.2 |
| 19. AE | 167 | 155 ± 156 | 1.9 | 177 | 141 | 150 ± 178 | 1.9 | 155 |
| 20. Va, km/s | 46.9 | 53 ± 26.8 | 0.31 | 44.7 | 47.1 | 54.8 ± 28.7 | 0.29 | 47.2 |
| 21. Vs, km/s | 57.4 | 51.1 ± 5.8 | 0.07 | 53.9 | 52.9 | 48.3 ± 4.1 | 0.04 | 51.6 |

[a] Last SW point before corresponding sequence of disturbed SW types. [b] First SW point after corresponding sequence of disturbed SW types.

**Table 4.** Average parameters for sequence **SW/SHEATH/EJECTA/SW (Figure 4).**

| Period | 1976–1996 | | | | | | 1997–2019 | | | | | |
|---|---|---|---|---|---|---|---|---|---|---|---|---|
| SW Type | SW [a] | SHEATH | | EJECTA | | SW [b] | SW [a] | SHEATH | | EJECTA | | SW [b] |
| Parameter | < > | < > ± σ | Stat. Err. | < > ± σ | Stat. Err. | < > | < > | < > ± σ | Stat. Err. | < > ± σ | Stat. Err. | < > |
| 1. N, cm$^{-3}$ | 9.3 | 10.5 ± 6.6 | 0.14 | 6.6 ± 4.6 | 0.07 | 9.1 | 8.3 | 8.6 ± 5.6 | 0.13 | 5.5 ± 4.0 | 0.07 | 7.4 |
| 2. Na/Np, % | 4.6 | 4.8 ± 3.1 | 0.08 | 5.1 ± 3.5 | 0.07 | 4.9 | 3.6 | 3.3 ± 2.1 | 0.06 | 3.0 ± 2.0 | 0.04 | 3.1 |
| 3. V, km/s | 411 | 448 ± 102 | 2.1 | 438 ± 94 | 1.4 | 415 | 401 | 431 ± 85 | 1.9 | 426 ± 80 | 1.4 | 408 |
| 4. Phi, deg | −0.9 | 0.7 ± 3.5 | 0.07 | 0.1 ± 2.9 | 0.04 | −0.1 | −0.7 | 0.8 ± 3.0 | 0.07 | −0.5 ± 2.6 | 0.05 | −0.7 |
| 5. Theta, deg | 1.1 | 1.3 ± 3.5 | 0.07 | 1.3 ± 3.1 | 0.05 | 1.2 | −0.2 | −0.4 ± 3.1 | 0.07 | −0.3 ± 2.3 | 0.04 | −0.6 |
| 6. T*10$^{-5}$, K | 0.80 | 1.70 ± 1.50 | 0.031 | 0.83 ± 0.68 | 0.010 | 1.09 | 0.70 | 1.42 ± 1.13 | 0.026 | 0.56 ± 0.53 | 0.009 | 0.73 |
| 7. T/Texp | 1.36 | 2.15 ± 1.07 | 0.022 | 1.18 ± 0.78 | 0.012 | 1.65 | 1.30 | 1.99 ± 1.06 | 0.024 | 0.83 ± 0.53 | 0.009 | 1.19 |
| 8. Ey, mV/m | 0.15 | 0.05 ± 2.07 | 0.043 | −0.02 ± 1.74 | 0.026 | 0.03 | 0.03 | 0.17 ± 1.63 | 0.037 | 0.02 ± 1.48 | 0.026 | 0.09 |
| 9. B, nT | 8.3 | 8.8 ± 3.8 | 0.08 | 7.3 ± 3.0 | 0.04 | 7.2 | 7.4 | 7.2 ± 3.1 | 0.07 | 6.3 ± 3.0 | 0.05 | 6.2 |
| 10. Bx, nT | −0.2 | 0.0 ± 4.8 | 0.10 | 0.0 ± 4.5 | 0.07 | 0.0 | −0.1 | −0.4 ± 3.5 | 0.08 | −0.3 ± 3.5 | 0.06 | −0.2 |
| 11. By, nT | 0.4 | 0.1 ± 5.4 | 0.11 | −0.2 ± 4.6 | 0.07 | 0.3 | −0.1 | 0.4 ± 4.6 | 0.10 | −0.2 ± 4.4 | 0.08 | 0.1 |
| 12. Bz, nT | −0.3 | −0.1 ± 4.5 | 0.09 | 0.1 ± 3.8 | 0.06 | −0.1 | −0.2 | −0.4 ± 3.7 | 0.08 | −0.1 ± 3.6 | 0.06 | −0.3 |
| 13. Pt*100, nPa | 0.9 | 2.2 ± 3.1 | 0.06 | 0.7 ± 0.8 | 0.01 | 1.1 | 0.7 | 1.4 ± 1.1 | 0.03 | 0.3 ± 0.3 | 0.01 | 0.6 |
| 14. Pd, nPa | 2.43 | 3.33 ± 2.53 | 0.052 | 1.95 ± 1.53 | 0.023 | 2.43 | 2.12 | 2.42 ± 1.51 | 0.035 | 1.52 ± 1.07 | 0.019 | 1.87 |
| 15. beta | 0.49 | 0.82 ± 0.79 | 0.017 | 0.41 ± 0.49 | 0.008 | 0.73 | 0.49 | 0.79 ± 0.71 | 0.016 | 0.32 ± 0.38 | 0.007 | 0.53 |
| 16. DST, nT | −17.4 | −19.8 ± 28.0 | 0.55 | −23.8 ± 25.6 | 0.36 | −18.6 | −12.7 | −14.0 ± 25.0 | 0.57 | −16.6 ± 20.4 | 0.36 | −13.7 |
| 17. DST*, nT | −20.6 | −26.7 ± 30.5 | 0.63 | −23.8 ± 26.5 | 0.39 | −20.1 | −14.4 | −17.5 ± 26.0 | 0.60 | −15.0 ± 21.4 | 0.38 | −12.9 |
| 18. Kp*10 | 27.2 | 31.9 ± 14.7 | 0.287 | 23.9 ± 13.5 | 0.188 | 25.8 | 22.75 | 25.0 ± 13.5 | 0.307 | 17.8 ± 12.90 | 0.227 | 19.8 |
| 19. AE | 283 | 310 ± 267 | 5.7 | 236 ± 223 | 3.5 | 256 | 214 | 236 ± 220 | 5.1 | 186 ± 209 | 3.7 | 203 |
| 20. Va, km/s | 61.1 | 60.3 ± 28.0 | 0.58 | 65.3 ± 32.1 | 0.48 | 55.5 | 55.1 | 55.8 ± 30.6 | 0.70 | 62.6 ± 34.5 | 0.61 | 51.2 |
| 21. Vs, km/s | 54.3 | 64.0 ± 13.7 | 0.29 | 54.5 ± 7.7 | 0.12 | 57.3 | 53.0 | 61.3 ± 11.6 | 0.27 | 51.1 ± 6.6 | 0.12 | 53.2 |

[a] Last SW point before corresponding sequence of disturbed SW types. [b] First SW point after corresponding sequence of disturbed SW types.

**Table 5.** Average parameters for sequence **SW/IS/SHEATH/EJECTA/SW (Figure 5).**

| Period | 1976–1996 | | | | | | 1997–2019 | | | | | |
|---|---|---|---|---|---|---|---|---|---|---|---|---|
| SW Type | SW [a] | SHEATH | | EJECTA | | SW [b] | SW [a] | SHEATH | | EJECTA | | SW [b] |
| Parameter | <> | <> ± σ | Stat. Err. | <> ± σ | Stat. Err. | <> | <> | <> ± σ | Stat. Err. | <> ± σ | Stat. Err. | <> |
| 1. N, cm$^{-3}$ | 11.2 | 14.8 ± 11.0 | 0.27 | 6.5 ± 4.6 | 0.09 | 7.8 | 9.0 | 11.5 ± 8.5 | 0.14 | 5.6 ± 4.5 | 0.06 | 7.0 |
| 2. Na/Np, % | 4.4 | 4.2 ± 2.6 | 0.08 | 4.4 ± 3.2 | 0.08 | 4.6 | 3.1 | 3.7 ± 2.3 | 0.05 | 3.9 ± 2.6 | 0.04 | 4.3 |
| 3. V, km/s | 396 | 446 ± 90 | 2.2 | 450 ± 84 | 1.6 | 440 | 400 | 464 ± 113 | 1.8 | 455 ± 107 | 1.3 | 439 |
| 4. Phi, deg | −0.6 | 0.1 ± 3.6 | 0.09 | 0.5 ± 3.1 | 0.06 | 0.2 | −1.6 | 0.0 ± 4.1 | 0.07 | −0.2 ± 2.8 | 0.04 | −0.7 |
| 5. Theta, deg | 1.0 | 1.1 ± 3.4 | 0.09 | 0.9 ± 3.0 | 0.06 | 1.2 | −0.5 | −0.4 ± 3.4 | 0.05 | −0.3 ± 2.6 | 0.03 | −0.4 |
| 6. T*10$^{-5}$, K | 0.82 | 1.72 ± 1.44 | 0.036 | 0.87 ± 0.66 | 0.012 | 1.25 | 0.73 | 1.80 ± 1.84 | 0.029 | 0.55 ± 0.67 | 0.008 | 0.83 |
| 7. T/Texp | 1.55 | 2.09 ± 1.06 | 0.026 | 1.12 ± 0.68 | 0.013 | 1.56 | 1.28 | 1.96 ± 1.13 | 0.018 | 0.70 ± 0.53 | 0.007 | 1.13 |
| 8. Ey, mV/m | −0.20 | −0.05 ± 2.59 | 0.062 | 0.12 ± 1.80 | 0.033 | 0.01 | 0.17 | 0.00 ± 2.80 | 0.044 | −0.03 ± 1.91 | 0.024 | 0.11 |
| 9. B, nT | 7.2 | 10.2 ± 4.4 | 0.11 | 7.5 ± 3.4 | 0.06 | 7.3 | 7.0 | 9.8 ± 4.8 | 0.08 | 7.3 ± 3.1 | 0.04 | 7.5 |
| 10. Bx, nT | −0.1 | −0.3 ± 4.8 | 0.12 | 0.0 ± 4.3 | 0.08 | −0.2 | 0.2 | 0.3 ± 4.6 | 0.07 | 0.2 ± 4.2 | 0.05 | 0.0 |
| 11. By, nT | −0.7 | −0.1 ± 6.6 | 0.16 | −0.4 ± 5.2 | 0.09 | −0.1 | 0.0 | 0.3 ± 6.4 | 0.10 | −0.3 ± 4.7 | 0.06 | 0.0 |
| 12. Bz, nT | 0.5 | 0.2 ± 5.4 | 0.13 | −0.2 ± 3.8 | 0.07 | 0.1 | −0.5 | 0.0 ± 5.4 | 0.09 | 0.0 ± 4.2 | 0.05 | −0.1 |
| 13. Pt*100, nPa | 1.0 | 3.2 ± 4.0 | 0.10 | 0.7 ± 0.7 | 0.01 | 1.2 | 0.7 | 2.4 ± 3.1 | 0.05 | 0.4 ± 0.6 | 0.01 | 0.7 |
| 14. Pd, nPa | 2.70 | 4.61 ± 3.75 | 0.090 | 2.10 ± 1.68 | 0.030 | 2.39 | 2.22 | 3.77 ± 2.83 | 0.045 | 1.78 ± 1.42 | 0.018 | 2.08 |
| 15. beta | 0.65 | 0.84 ± 0.89 | 0.022 | 0.44 ± 0.61 | 0.012 | 0.72 | 0.52 | 0.75 ± 0.83 | 0.013 | 0.24 ± 0.35 | 0.004 | 0.38 |
| 16. DST, nT | −8.9 | −16.2 ± 27.9 | 0.65 | −27.9 ± 24.3 | 0.42 | −26.1 | −10.5 | −20.5 ± 38.7 | 0.61 | −25.4 ± 27.1 | 0.34 | −20.9 |
| 17. DST*, nT | −11.0 | −26.8 ± 28.0 | 0.67 | −29.3 ± 26.4 | 0.48 | −27.4 | −10.2 | −28.8 ± 41.7 | 0.66 | −25.1 ± 27.6 | 0.35 | −20.6 |
| 18. Kp*10 | 25.5 | 35.3 ± 14.2 | 0.330 | 26.9 ± 14.6 | 0.254 | 28.6 | 26.88 | 32.3 ± 16.0 | 0.254 | 23.1 ± 14.9 | 0.188 | 25.1 |
| 19. AE | 219 | 335 ± 278 | 6.9 | 264 ± 235 | 4.4 | 230 | 222 | 323 ± 288 | 4.6 | 245 ± 252 | 3.2 | 272 |
| 20. Va, km/s | 49.7 | 61.4 ± 29.9 | 0.72 | 67.2 ± 32.6 | 0.59 | 60.3 | 52.2 | 68.3 ± 47.5 | 0.76 | 74.2 ± 44.8 | 0.56 | 62.7 |
| 21. Vs, km/s | 54.3 | 64.3 ± 14.0 | 0.35 | 55.1 ± 7.9 | 0.15 | 59.3 | 53.2 | 64.6 ± 15.9 | 0.25 | 50.8 ± 7.5 | 0.09 | 54.4 |

[a] Last SW point before corresponding sequence of disturbed SW types. [b] First SW point after corresponding sequence of disturbed SW types.

Overall, Figures 3–5 show that in sheath and ejecta the same trend is observed as in CIR: in the epoch of low solar activity, the time profiles have the same shape, but pass at lower levels than in the epoch of high activity. There is also a positive latitudinal angle at high activity and a negative angle at low activity.

However, some differences are observed. In contrast to CIR, in ejecta, in the epoch of low activity, the values of the dimensionless parameters $\beta$ and T/Texp noticeably decrease, while in sheath their changes are small, similarly to CIR. As we noted earlier [23], the change in longitude angle in sheath is similar to its change in CIR; however, unlike CIR, in sheath, its amplitude does not noticeably change when moving from high to low activity.

### 3.3. Variation in Sheath and MC Events

Unlike the previous section, in this section we will consider magnetic clouds instead of ejecta. Figures 6–8 show the timing profiles for the sequences SW/MC/SW, SW/sheath/MC/SW, and SW/IS/sheath/MC/SW, respectively. The average values of the parameters are presented in Tables 6–8.

It should be noted that the strong variability of time profiles for parameters that include MC data (Figures 6–8) is associated with low statistics of observation cases (see figure captions). Nevertheless, these smoothed profiles make it possible to judge the general trend in the change in profiles during the transition from high to low solar activity. In particular, the data presented demonstrate that the profiles for MC-related events behave similarly to the profiles for ejecta and have the same features: in the epoch of low solar activity, the time profiles have the similar shape, but pass at lower levels than in the epoch of high activity. There is also a positive latitudinal angle at high activity and a negative angle at low activity. We also note that the previously discovered anticorrelation of the parameters $\beta$ and Na/Np [24] is visible only for the MC and for the epoch of low activity. This is discussed in more detail in an article in this Special Issue of the journal [29].

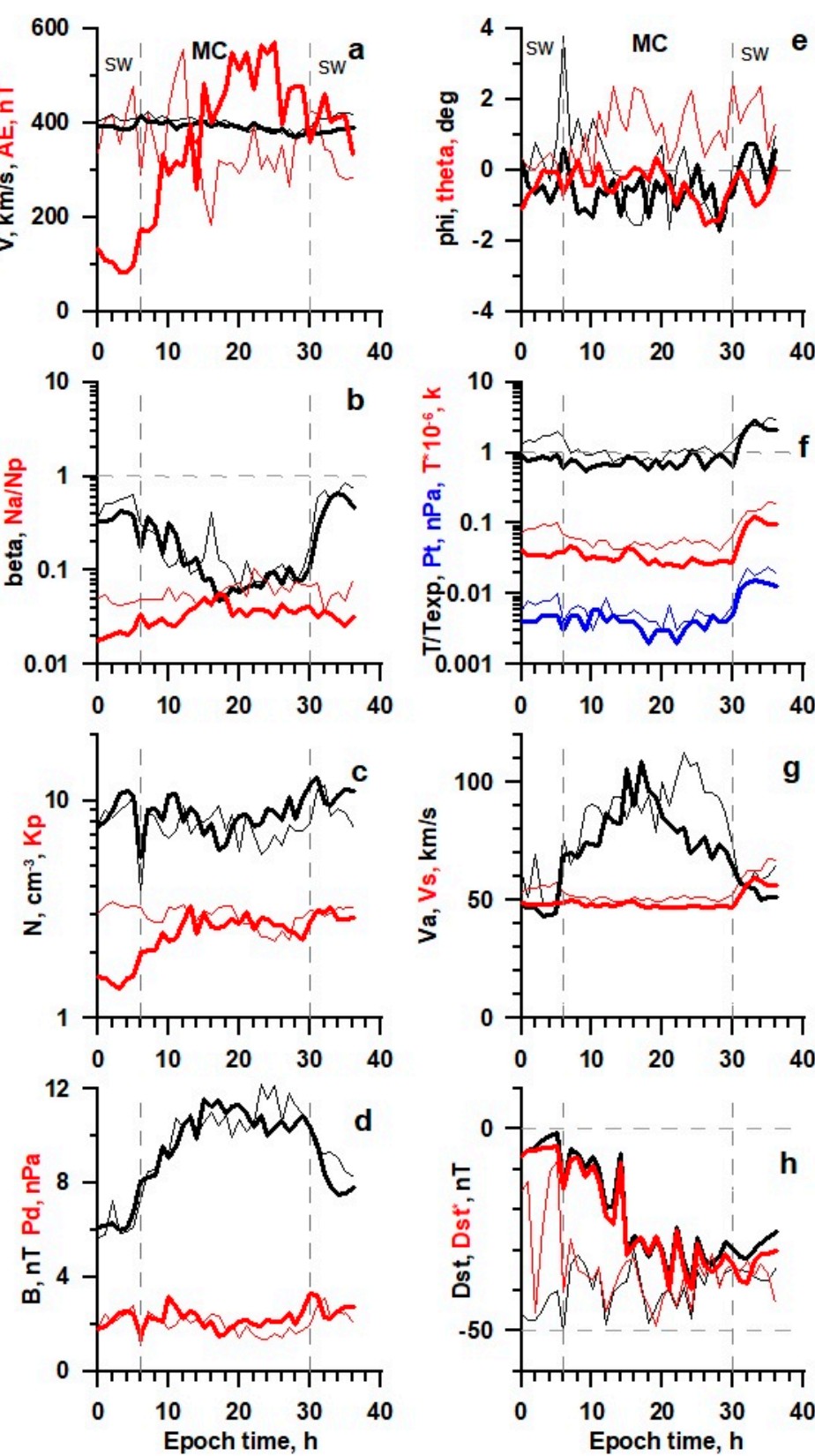

**Figure 6.** The temporal profile of solar wind and IMF parameters (see legend in the text) for MC obtained by the double superposed epoch analysis for 13 events in 21–22 SCs (thin lines) and 13 events in 23–24 SCs (thick lines).

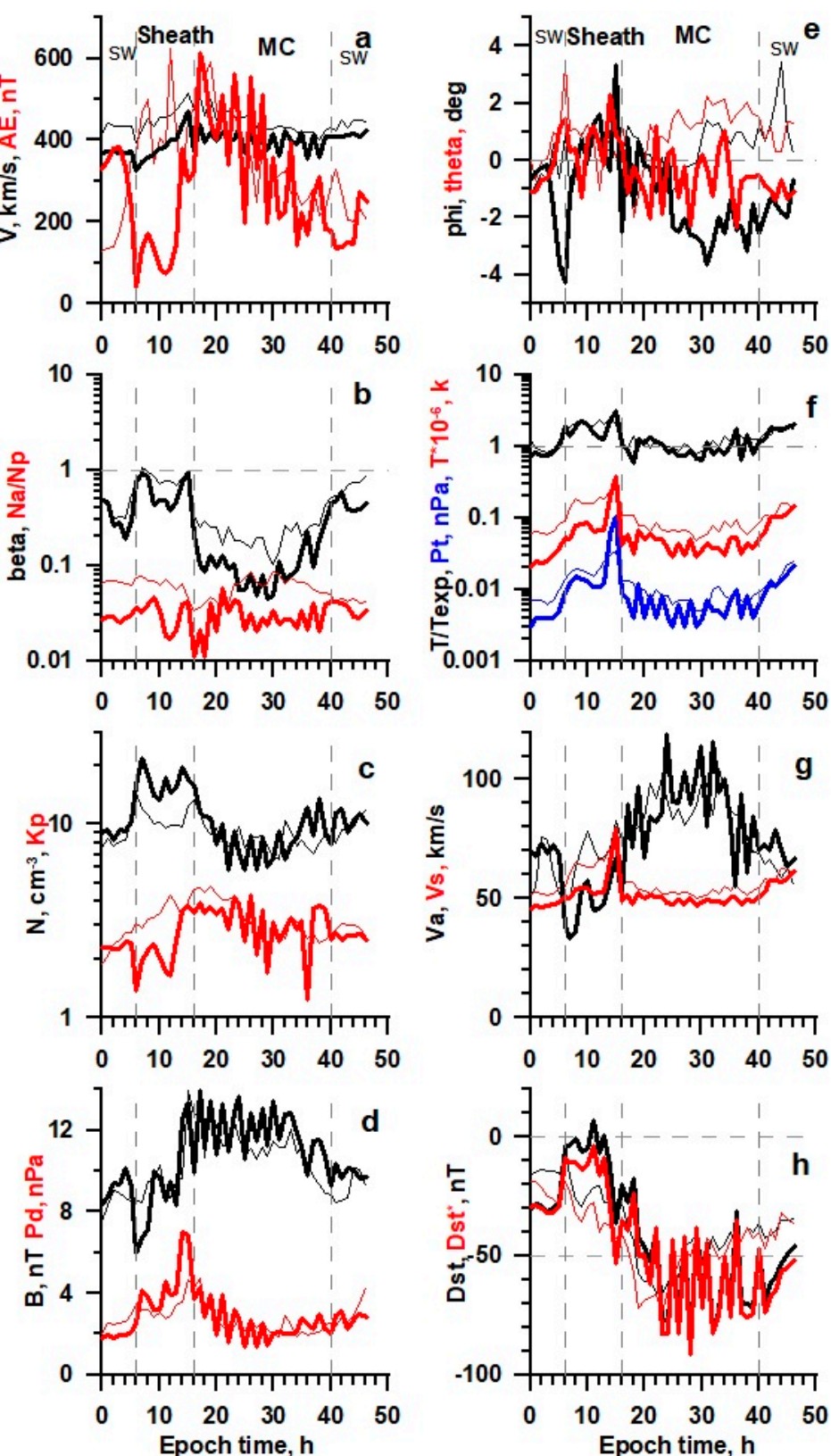

**Figure 7.** The temporal profile of solar wind and IMF parameters (see legend in the text) for Sheath/MC obtained by the double superposed epoch analysis for 15 events in 21–22 SCs (thin lines) and 6 events in 23–24 SCs (thick lines).

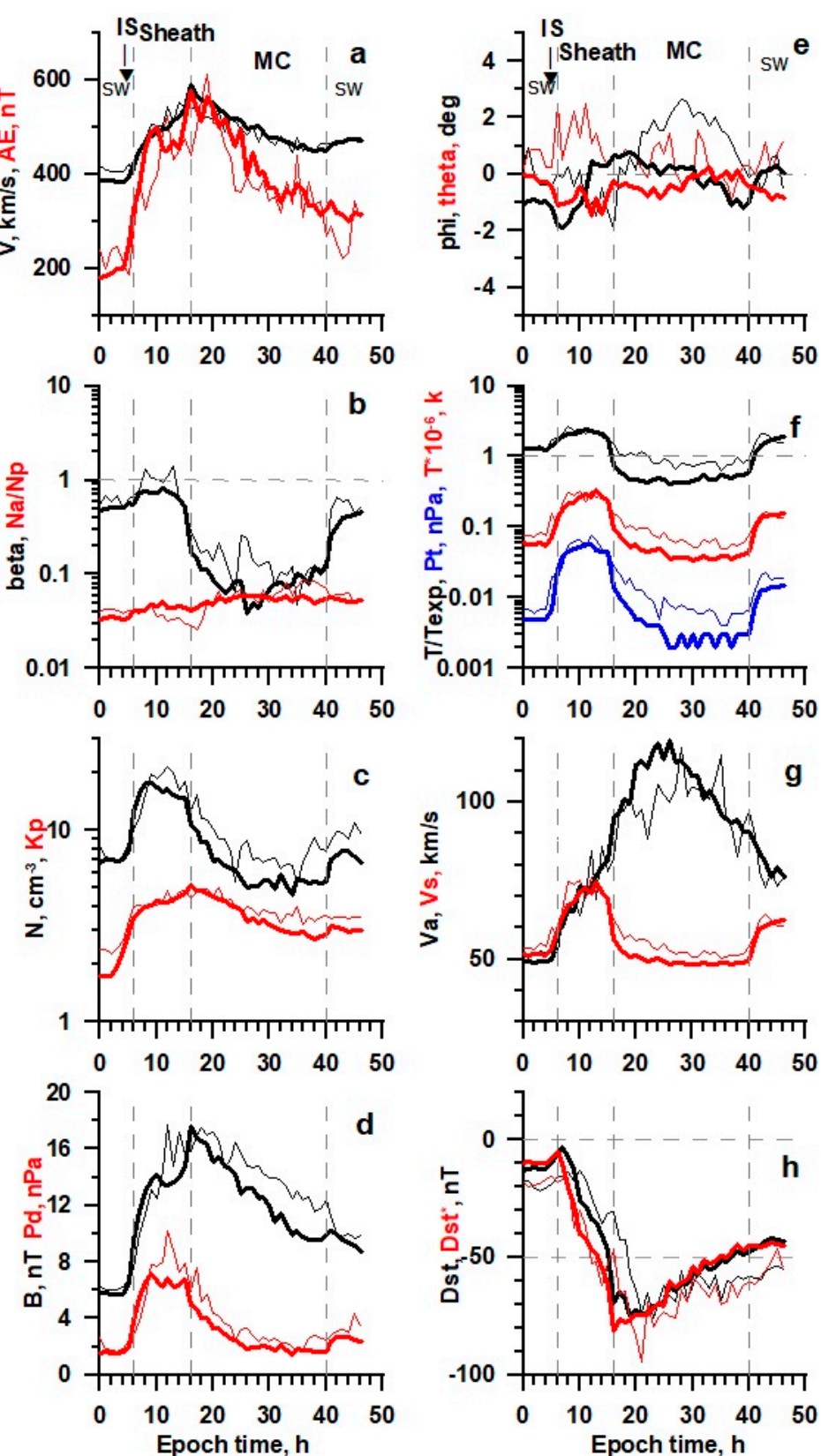

**Figure 8.** The temporal profile of solar wind and IMF parameters (see legend in the text) for IS/Sheath/Ejecta obtained by the double superposed epoch analysis for 32 events in 21–22 SCs (thin lines) and 123 events in 23–24 SCs (thick lines).

**Table 6.** Average parameters for sequence **SW/MC/SW (Figure 6).**

| Period | 1976–1996 | | | | 1997–2019 | | | |
|---|---|---|---|---|---|---|---|---|
| SW Type | SW [a] | MC | | SW [b] | SW [a] | MC | | SW [b] |
| Parameter | <> | <> ± σ | Stat. Err. | <> | <> | <> ± σ | Stat. Err. | <> |
| 1. N, cm$^{-3}$ | 10.4 | 7.2 ± 6.2 | 0.34 | 11.3 | 10.6 | 8.7 ± 6.0 | 0.33 | 12.8 |
| 2. Na/Np, % | 4.6 | 6.1 ± 4.6 | 0.34 | 7.3 | 2.3 | 3.8 ± 3.0 | 0.20 | 3.8 |
| 3. V, km/s | 410 | 400 ± 57 | 3.1 | 410 | 389 | 391 ± 61 | 3.4 | 378 |
| 4. Phi, deg | 1.0 | −0.2 ± 3.4 | 0.19 | 0.0 | −0.5 | −0.7 ± 2.2 | 0.12 | 0.2 |
| 5. Theta, deg | 0.2 | 1.2 ± 3.5 | 0.20 | 1.3 | 0.0 | −0.5 ± 2.3 | 0.13 | 0.0 |
| 6. T*10$^{-5}$, K | 1.03 | 0.53 ± 0.41 | 0.022 | 1.10 | 0.39 | 0.32 ± 0.25 | 0.014 | 0.60 |
| 7. T/Texp | 2.05 | 1.00 ± 0.66 | 0.036 | 1.83 | 0.93 | 0.75 ± 0.81 | 0.045 | 1.49 |
| 8. Ey, mV/m | −0.52 | 0.45 ± 2.44 | 0.133 | −0.19 | −0.54 | 0.80 ± 2.43 | 0.135 | 0.51 |
| 9. B, nT | 6.1 | 10.5 ± 4.0 | 0.22 | 9.4 | 6.8 | 10.3 ± 3.5 | 0.19 | 9.6 |
| 10. Bx, nT | −0.1 | −2.0 ± 4.5 | 0.24 | −2.5 | 0.6 | 0.5 ± 4.6 | 0.26 | −1.6 |
| 11. By, nT | 1.9 | 1.6 ± 7.4 | 0.40 | 2.0 | −1.1 | 0.9 ± 7.1 | 0.39 | −2.9 |
| 12. Bz, nT | 1.4 | −1.3 ± 6.1 | 0.33 | 0.9 | 1.2 | −2.1 ± 6.0 | 0.33 | −1.4 |
| 13. Pt*100, nPa | 1.0 | 0.5 ± 0.6 | 0.03 | 1.5 | 0.5 | 0.4 ± 0.5 | 0.03 | 1.1 |
| 14. Pd, nPa | 2.79 | 1.91 ± 1.58 | 0.084 | 2.81 | 2.43 | 2.23 ± 1.75 | 0.097 | 3.22 |
| 15. beta | 0.63 | 0.14 ± 0.25 | 0.014 | 0.60 | 0.39 | 0.13 ± 0.21 | 0.012 | 0.27 |
| 16. DST, nT | −39.9 | −38.8 ± 35.4 | 1.82 | −34.8 | −0.8 | −23.6 ± 26.7 | 1.48 | −31.5 |
| 17. DST*, nT | −7.9 | −37.7 ± 34.9 | 1.87 | −35.4 | −4.1 | −25.3 ± 27.7 | 1.54 | −37.5 |
| 18. Kp*10 | 33.00 | 28.8 ± 15.3 | 0.787 | 30.9 | 15.6 | 26.0 ± 15.7 | 0.872 | 31.2 |
| 19. AE | 479 | 350 ± 345 | 21.3 | 420 | 96 | 412 ± 336 | 22.4 | 406 |
| 20. Va, km/s | 50.3 | 93.9 ± 42.7 | 2.32 | 56.4 | 44.4 | 80.0 ± 33.9 | 1.88 | 57.4 |
| 21. Vs, km/s | 57.4 | 50.8 ± 5.4 | 0.29 | 58.1 | 49.0 | 47.9 ± 3.4 | 0.19 | 51.8 |

[a] Last SW point before corresponding sequence of disturbed SW types. [b] First SW point after corresponding sequence of disturbed SW types.

**Table 7.** Average parameters for sequence **SW/SHEATH/MC/SW (Figure 7).**

| Period | 1976–1996 | | | | | | 1997–2019 | | | | | |
|---|---|---|---|---|---|---|---|---|---|---|---|---|
| SW Type | SW [a] | SHEATH | | MC | | SW [b] | SW [a] | SHEATH | | MC | | SW [b] |
| Parameter | <> | <> ± σ | Stat. Err. | <> ± σ | Stat. Err. | <> | <> | <> ± σ | Stat. Err. | <> ± σ | Stat. Err. | <> |
| 1. N, cm$^{-3}$ | 10.8 | 11.1 ± 7.9 | 0.64 | 8.8 ± 6.4 | 0.30 | 8.0 | 10.5 | 17.0 ± 8.1 | 0.82 | 8.9 ± 5.3 | 0.47 | 11.3 |
| 2. Na/Np, % | 6.2 | 6.5 ± 3.9 | 0.41 | 5.9 ± 4.3 | 0.27 | 4.2 | 3.0 | 3.2 ± 2.1 | 0.23 | 3.0 ± 2.9 | 0.28 | 4.1 |
| 3. V, km/s | 434 | 459 ± 94 | 7.6 | 441 ± 84 | 4.0 | 424 | 371 | 391 ± 70 | 7.1 | 400 ± 70 | 6.2 | 410 |
| 4. Phi, deg | −0.6 | 0.5 ± 3.3 | 0.27 | 0.4 ± 2.9 | 0.14 | 0.7 | −3.6 | 0.3 ± 3.8 | 0.38 | −1.7 ± 2.2 | 0.19 | −1.9 |
| 5. Theta, deg | 0.8 | 0.8 ± 3.2 | 0.26 | 1.2 ± 3.4 | 0.17 | 0.8 | 1.1 | 0.6 ± 2.7 | 0.27 | −0.5 ± 2.4 | 0.21 | −0.9 |
| 6. T*10$^{-5}$, K | 0.82 | 1.77 ± 1.27 | 0.103 | 0.73 ± 0.60 | 0.028 | 1.10 | 0.37 | 1.20 ± 1.79 | 0.181 | 0.43 ± 0.25 | 0.022 | 0.65 |
| 7. T/Texp | 1.17 | 2.04 ± 0.93 | 0.075 | 1.10 ± 0.88 | 0.042 | 1.48 | 0.99 | 1.91 ± 1.30 | 0.131 | 0.95 ± 0.71 | 0.063 | 1.26 |
| 8. Ey, mV/m | 0.03 | 0.33 ± 2.34 | 0.190 | 0.70 ± 3.11 | 0.147 | −0.08 | −1.20 | −0.84 ± 2.62 | 0.265 | 1.69 ± 3.05 | 0.270 | 0.21 |
| 9. B, nT | 8.5 | 10.0 ± 5.1 | 0.41 | 11.0 ± 3.6 | 0.17 | 8.5 | 9.2 | 9.6 ± 5.2 | 0.53 | 12.0 ± 4.5 | 0.40 | 10.4 |
| 10. Bx, nT | 3.0 | 0.1 ± 6.0 | 0.49 | −1.8 ± 5.3 | 0.25 | 0.6 | 0.4 | −0.6 ± 4.0 | 0.40 | −1.0 ± 7.0 | 0.62 | 2.1 |
| 11. By, nT | −2.7 | −0.1 ± 6.7 | 0.54 | 1.6 ± 6.6 | 0.31 | 2.2 | 0.3 | 0.6 ± 6.4 | 0.65 | 0.6 ± 7.0 | 0.62 | 0.8 |
| 12. Bz, nT | −0.4 | −0.9 ± 4.7 | 0.38 | −1.3 ± 6.8 | 0.32 | 0.3 | 3.0 | 2.1 ± 5.8 | 0.59 | −3.8 ± 6.3 | 0.56 | −0.4 |
| 13. Pt*100, nPa | 1.0 | 2.2 ± 2.0 | 0.16 | 0.8 ± 0.8 | 0.04 | 1.1 | 0.5 | 2.9 ± 5.7 | 0.58 | 0.6 ± 0.6 | 0.05 | 0.9 |
| 14. Pd, nPa | 2.98 | 3.41 ± 2.03 | 0.164 | 2.55 ± 1.60 | 0.075 | 2.38 | 2.13 | 4.52 ± 3.27 | 0.330 | 2.55 ± 2.43 | 0.215 | 2.91 |
| 15. beta | 0.55 | 0.80 ± 0.82 | 0.067 | 0.24 ± 0.32 | 0.015 | 0.53 | 0.29 | 0.64 ± 0.48 | 0.048 | 0.12 ± 0.18 | 0.016 | 0.46 |
| 16. DST, nT | −15.9 | −23.8 ± 22.9 | 1.72 | −48.6 ± 40.0 | 1.77 | −42.7 | −27.2 | −8.1 ± 31.5 | 3.18 | −57.6 ± 49.4 | 4.37 | −68.7 |
| 17. DST*, nT | −28.4 | −32.7 ± 21.4 | 1.74 | −51.8 ± 39.1 | 1.84 | −46.7 | −28.6 | −18.5 ± 31.8 | 3.21 | −61.0 ± 55.5 | 4.91 | −73.5 |
| 18. Kp*10 | 27.5 | 35.7 ± 14.8 | 1.109 | 33.2 ± 18.8 | 0.833 | 28.7 | 24.50 | 24.3 ± 16.3 | 1.651 | 32.7 ± 21.3 | 1.878 | 27.7 |
| 19. AE | 362 | 420 ± 333 | 29.1 | 362 ± 313 | 15.8 | 330 | 235 | 176 ± 229 | 23.9 | 351 ± 315 | 30.2 | 135 |
| 20. Va, km/s | 52.0 | 71.9 ± 43.4 | 3.53 | 88.7 ± 43.1 | 2.04 | 64.3 | 68.1 | 51.8 ± 31.8 | 3.21 | 90.6 ± 40.2 | 3.55 | 72.6 |
| 21. Vs, km/s | 54.5 | 65.0 ± 12.9 | 1.05 | 53.3 ± 7.3 | 0.34 | 58.1 | 48.7 | 57.5 ± 16.4 | 1.66 | 49.5 ± 3.5 | 0.31 | 52.7 |

[a] Last SW point before corresponding sequence of disturbed SW types. [b] First SW point after corresponding sequence of disturbed SW types.

**Table 8.** Average parameters for sequence **SW/IS/SHEATH/MC/SW (Figure 8).**

| Period | 1976–1996 | | | | | | 1997–2019 | | | | | |
|---|---|---|---|---|---|---|---|---|---|---|---|---|
| SW Type | SW [a] | SHEATH | | MC | | SW [b] | SW [a] | SHEATH | | MC | | SW [b] |
| Parameter | <> | <> ± σ | Stat. Err. | <> ± σ | Stat. Err. | <> | <> | <> ± σ | Stat. Err. | <> ± σ | Stat. Err. | <> |
| 1. N, cm$^{-3}$ | 8.5 | 17.4 ± 11.5 | 0.73 | 8.7 ± 6.9 | 0.28 | 9.0 | 8.0 | 15.9 ± 11.1 | 0.28 | 6.2 ± 6.2 | 0.11 | 7.1 |
| 2. Na/Np, % | 4.1 | 3.7 ± 2.6 | 0.22 | 6.1 ± 4.9 | 0.24 | 5.6 | 3.4 | 4.5 ± 3.4 | 0.10 | 5.4 ± 3.6 | 0.08 | 5.6 |
| 3. V, km/s | 422 | 495 ± 145 | 9.2 | 477 ± 116 | 4.6 | 465 | 392 | 491 ± 138 | 3.5 | 496 ± 123 | 2.1 | 463 |
| 4. Phi, deg | −0.4 | −0.6 ± 3.3 | 0.21 | 1.3 ± 3.7 | 0.15 | 0.0 | −1.2 | −0.6 ± 4.0 | 0.10 | −0.1 ± 3.2 | 0.06 | −0.3 |
| 5. Theta, deg | 0.2 | 1.2 ± 4.1 | 0.27 | 0.4 ± 3.8 | 0.16 | 0.0 | −0.4 | −0.9 ± 4.2 | 0.11 | −0.3 ± 2.8 | 0.05 | −0.5 |
| 6. T*10$^{-5}$, K | 1.55 | 2.63 ± 3.15 | 0.201 | 0.75 ± 0.70 | 0.028 | 1.17 | 0.74 | 2.44 ± 3.72 | 0.094 | 0.45 ± 0.46 | 0.008 | 0.98 |
| 7. T/Texp | 1.70 | 2.16 ± 1.18 | 0.075 | 0.89 ± 0.67 | 0.027 | 1.56 | 1.44 | 2.09 ± 1.67 | 0.042 | 0.50 ± 0.47 | 0.008 | 1.12 |
| 8. Ey, mV/m | 0.07 | −0.19 ± 4.55 | 0.271 | 0.31 ± 5.34 | 0.200 | −0.72 | 0.15 | 0.33 ± 4.52 | 0.114 | 0.47 ± 4.70 | 0.081 | −0.15 |
| 9. B, nT | 6.8 | 13.7 ± 7.2 | 0.42 | 14.4 ± 6.2 | 0.22 | 11.2 | 6.5 | 13.3 ± 7.8 | 0.20 | 12.6 ± 6.8 | 0.12 | 10.2 |
| 10. Bx, nT | −0.1 | 0.7 ± 6.0 | 0.35 | 1.2 ± 6.6 | 0.24 | 0.5 | −0.4 | 0.3 ± 6.3 | 0.16 | 0.3 ± 6.8 | 0.12 | 0.4 |
| 11. By, nT | 0.1 | 1.4 ± 9.5 | 0.56 | −0.7 ± 9.6 | 0.35 | 0.9 | 0.0 | 1.3 ± 9.0 | 0.23 | 0.7 ± 8.5 | 0.15 | −0.2 |
| 12. Bz, nT | 0.0 | 0.8 ± 7.5 | 0.44 | −0.2 ± 9.7 | 0.35 | 1.8 | −0.3 | −0.6 ± 7.6 | 0.19 | −1.1 ± 8.6 | 0.15 | 0.2 |
| 13. Pt*100, nPa | 2.0 | 5.2 ± 5.4 | 0.34 | 0.9 ± 1.1 | 0.04 | 1.3 | 0.7 | 4.6 ± 8.6 | 0.22 | 0.4 ± 0.7 | 0.01 | 0.8 |
| 14. Pd, nPa | 2.45 | 7.14 ± 5.19 | 0.307 | 3.28 ± 3.77 | 0.140 | 2.76 | 1.98 | 6.32 ± 6.35 | 0.161 | 2.42 ± 2.66 | 0.046 | 2.38 |
| 15. beta | 0.64 | 0.93 ± 1.15 | 0.074 | 0.14 ± 0.24 | 0.010 | 0.47 | 0.56 | 0.69 ± 0.73 | 0.018 | 0.09 ± 0.19 | 0.003 | 0.25 |
| 16. DST, nT | −18.5 | −22.3 ± 40.7 | 2.35 | −60.6 ± 51.6 | 1.76 | −59.0 | −10.3 | −25.0 ± 54.6 | 1.37 | −60.7 ± 51.1 | 0.88 | −46.3 |
| 17. DST*, nT | −15.7 | −39.1 ± 47.3 | 2.80. | −68.7 ± 53.6 | 1.99 | −61.0 | −8.2 | −37.1 ± 55.5 | 1.40 | −60.8 ± 50.1 | 0.86 | −44.9 |
| 18. Kp*10 | 34.0 | 43.4 ± 17.1 | 0.987 | 39.8 ± 19.3 | 0.659 | 34.8 | 29.7 | 42.0 ± 19.3 | 0.482 | 35.8 ± 20.6 | 0.353 | 31.5 |
| 19. AE | 188 | 419 ± 358 | 21.5 | 401 ± 319 | 11.3 | 291 | 253 | 454 ± 396 | 10.0 | 424 ± 361 | 6.3 | 343 |
| 20. Va, km/s | 52.7 | 74.9 ± 45.1 | 2.69 | 126.6 ± 78.2 | 2.93 | 89.6 | 49.8 | 79.0 ± 57.2 | 1.45 | 128.8 ± 79.9 | 1.38 | 86.7 |
| 21. Vs, km/s | 59.9 | 71.0 ± 24.3 | 1.55 | 53.4 ± 7.9 | 0.32 | 58.4 | 53.4 | 69.0 ± 24.8 | 0.63 | 49.6 ± 5.7 | 0.10 | 56.2 |

[a] Last SW point before corresponding sequence of disturbed SW types. [b] First SW point after corresponding sequence of disturbed SW types.

Thus, for all parameters and for all sequences of SW types in Figures 1–8, the average profiles for 21–22 (thin lines) and 23–24 (thick lines) SC have similar shapes, but the parameters for 23–24 SC have lower values (including both the magnitudes of the measured and density-corrected Dst and Dst* indices, i.e., reflecting lower ring current activity).

The only parameters that changed in a different way were the mean angles of arrival of the solar wind stream, longitude $\phi$ and latitude $\theta$ in panels (e) of Figures 1–8. Firstly, the average latitudinal angle $\theta$ at the time of 21–22 SC had a small positive value (~0.5–1.0 degrees), and at the time of 23–24 SC it had the same small negative value. Secondly, in contrast to other SW parameters, the average longitude angle $\phi$ in the interaction regions of the CIR types (with and without an interplanetary shock wave) varied in a wider range (it was larger in absolute value) by 23–24 SC than for the corresponding types SW at 21–22 SC.

## 4. Discussion and Conclusions

Thus, we studied the time profiles of the main parameters of SW and IMF for the eight usual sequences of SW phenomena: (1) SW/CIR/SW, (2) SW/IS/CIR/SW, (3) SW/ejecta/SW, (4) SW/sheath/ejecta/SW, (5) SW/IS/sheath/ejecta/SW, (6) SW/MC/SW, (7) SW/sheath/MC/SW, and (8) SW/IS/sheath/MC/SW and compared time profiles for epochs high (21–22 solar cycles) and low (23–24 CS) solar activity.

Since the periods of observations in the epoch of high solar activity (1976–1996) and the period covering the observations of our previous work [21] (1976–2000) coincide significantly, the time profiles for the period of high activity in Figures 1–8 (thin lines) almost coincide with the profiles presented in the previous work for a close time interval. Time profiles for the period of low solar activity (thick lines in Figures 1–8) have a similar shape, but are located at lower values. Both these profiles and the average values in Tables 1–8 are in good agreement with the data averaged over cycle phases and intervals of SW types [21]. Thus, the obtained results indicate that a decrease in SW parameters in the epoch of low solar activity is observed on time scales smaller than the sizes of SW

phenomena and manifests itself in a shift of time profiles characteristic of the previous epoch towards lower values.

It should be noted that although the shapes of the time profiles of the parameters for the epochs of high and low solar activity are similar, the difference in amplitude for the parameters is different. Changes in the undisturbed SW streams at the beginning and end of Figures 1–8 and in Tables 1–8 are also distinguished from changes in the disturbed streams CIRs, sheaths, MCs and ejecta. Apparently, this is due to the fact that the properties of undisturbed SW flows depend on the properties of the solar corona and their changes in different epochs of solar activity, while the properties of disturbed SW types, in addition to the properties of the corona, additionally depend on the changed conditions of interacting flows and differences in formation processes of disturbed SW streams. The properties of the magnetosphere can additionally depend on the change in the nature of the interaction of the solar wind with the magnetosphere with a decrease in the values of the main parameters of the solar wind, in particular, an increase in the sound and Alfven Mach numbers [19].

Based on the time profiles of SW parameters for the interval 1976–2000, we previously have assumed [22] that the formation of different pairs of SW types (both types of compression regions, CIRs and sheaths, and both types of ICMEs, MCs and ejecta) may occur due to the same mechanisms, and differences between them (CIRs vs. sheaths and MCs vs. ejecta) may arise due to differences in the geometry of observations, in particular, due to differences in the angle between the normal to the plane of the piston (or high-speed wind, or ICME) and the trajectory satellite. Since the shapes of time profiles during the epoch of low solar activity did not change significantly respect with the epoch oh high activity, but only shifted towards lower values, the assumption made earlier remains valid. The fact that the amplitude of the change in the longitudinal angle increased during the epoch of low activity for CIR, but remained unchanged for sheath, can also be explained in terms of this hypothesis. The detected small shift of the mean latitudinal angle to positive values for the epoch of high solar activity and to negative values during the epoch of low solar activity is obviously related to the north–south asymmetry of the activity of the solar corona, which cannot be explained with variations in even–odd sequences of solar cycles, since averaging was carried out in adjacent cycles.

**Author Contributions:** Conceptualization, Y.I.Y.; methodology, Y.I.Y.; software, I.G.L. and A.A.K.; validation, M.Y.Y., M.O.R. and L.S.R.; formal analysis, N.L.B.; investigation, O.V.S.; resources, A.V.M.; data curation, I.G.L.; writing—original draft preparation, Y.I.Y.; writing—review and editing, Y.I.Y.; visualization, I.G.L.; supervision, Y.I.Y. All authors have read and agreed to the published version of the manuscript.

**Funding:** The work was supported by the Russian Science Foundation, grant 22-12-00227.

**Acknowledgments:** Authors thank creators of databases https://spdf.gsfc.nasa.gov/pub/data/omni/low_res_omni (1 November 2021)and http://www.iki.rssi.ru/pub/omni (1 January 2022) for possibility to use in the work.

**Conflicts of Interest:** The authors declare no conflict of interest.

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
