# Peer review of "Dynamics of Large-Scale Solar-Wind Streams Obtained by the Double Superposed Epoch Analysis: 5. Influence of the Solar Activity Decrease"

_universe, doi:10.3390/universe8090472_

Round 1
Reviewer 1 Report
The paper entitled “Dynamics of Large‐Scale Solar‐Wind Streams Obtained by the Double Superposed Epoch Analysis: 5. Influence of the Solar Activity Decrease” presents a detailed analysis of the effect of low and high solar activities on the solar wind, CIR, and other parameters using the superposed epoch analysis. I have the following comments and suggestions for the authors which must be taken into account before the work can be accepted for publication.
1. While the analysis is statistically significant, the results derived from them were not discussed sufficiently enough, and some have not been discussed at all, e.g. Fig 1 & 2 & respective tables.
2. In general, low activity duration resulted in low values for all the parameters estimated within the epoch analysis, and the same has been indicated in the discussion. Nevertheless, the difference in amplitude for the parameters is different. There should be a discussion dedicated to all the individual parameters presented in figures and tables signifying the difference and its relation with solar activity.
3. Although the trend of the parameters is the same in both the low and high activity, the variation of the difference between the investigated parameters on the time axis is not uniform. For example, in figure 1, between the vertical dotted lines representing CIR, the relative variation between the parameters is different than that during SW. A similar effect can be seen in the parameters namely AE, N, B, and Dst in figure 2. I failed to find a discussion on this and its interpretation in the manuscript. It will be interesting to see a correlation of the differences with a quantifier of “high” and “low” solar activity periods, e.g. with the peak sunspot number in a solar cycle.
4. Authors have inferred the differences in the plane of the piston and normal to be the reason for lower values during the low activity period. More elaboration is required to establish the systematically lower shift in the values of the parameters due to this.
5. Statistical errors may be shown in the individual figure in the form of error bars as a legend for the reader to appreciate the statistical significance of the data.
6. A minor comment=> Replace “Thing Line” with “Thin Line” in figure captions and manuscript text.
Reviewer 2 Report
This is a continuous work of the authors' effort to develop temporal average properties of several types of SW sequences in particular to compare low and high solar activity phases. The information contained in the manuscript is useful to report to the community. Overall the manuscript is well written and alomost ready for publication except for the following very minor points which can be corrected without rereview by this referee.
- OMNI base --> OMNI database?
- Ey in all Tables: must be y component of electric field, so its unit should be someting like mV/m rather than nT?
- Line 271: "more low values" -- did you mean "lower values"?
